# Proinflammatory Endothelial Phenotype in Very Preterm Infants: A Pilot Study

**DOI:** 10.3390/biomedicines10051185

**Published:** 2022-05-20

**Authors:** Giacomo S. Amelio, Livia Provitera, Genny Raffaeli, Ilaria Amodeo, Silvia Gulden, Valeria Cortesi, Francesca Manzoni, Nicola Pesenti, Matteo Tripodi, Valentina Pravatà, Caterina Lonati, Gaia Cervellini, Fabio Mosca, Giacomo Cavallaro

**Affiliations:** 1Neonatal Intensive Care Unit, Fondazione IRCCS Ca’ Granda Ospedale Maggiore Policlinico, 20122 Milan, Italy; giacomo.amelio3@gmail.com (G.S.A.); livia.provitera@gmail.com (L.P.); genny.raffaeli@unimi.it (G.R.); amodeoilaria@gmail.com (I.A.); silvia.gulden@hotmail.it (S.G.); valeria.cortesi@unimi.it (V.C.); francescamanzoni.unimi@gmail.com (F.M.); nicolapesenti@hotmail.it (N.P.); matteo.tripodi.mt@gmail.com (M.T.); valentinapravata94@gmail.com (V.P.); gaia.cervellini@unimi.it (G.C.); fabio.mosca@unimi.it (F.M.); 2Department of Clinical Sciences and Community Health, Università degli Studi di Milano, 20122 Milan, Italy; 3Department of Statistics and Quantitative Methods, Division of Biostatistics, Epidemiology, and Public Health, University of Milano-Bicocca, 20126 Milan, Italy; 4Center for Preclinical Investigation, Fondazione IRCCS Ca’ Granda Ospedale Maggiore Policlinico, 20122 Milan, Italy; caterina.lonati@policlinico.mi.it

**Keywords:** inflammation, angiogenesis, angiopoietins, adhesion molecules, endothelium, newborn, infant

## Abstract

Very preterm infants are exposed to prenatal inflammatory processes and early postnatal hemodynamic and respiratory complications, but limited data are available about the endothelial effect of these conditions. The present pilot study investigates the perinatal endothelial phenotype in very preterm infants (VPIs) and explores its predictive value on neonatal mortality and hemodynamic and respiratory complications. Angiopoietin 1 (Ang-1), Ang-2, E-selectin, vascular adhesion molecule 1 (VCAM-1), tissue factor (TF), and endothelin 1 (ET-1) concentrations were tested in first (T1), 3rd (T2), and 7–10th (T3) day of life in 20 VPIs using Luminex technology and compared with 14 healthy full-term infants (FTIs). Compared to FTIs, VPIs had lower Ang-1 at T1 and T2; higher Ang-2 at T1, T2, and T3; higher Ang-2/Ang-1 ratio at T1, T2, and T3; lower E-selectin at T1, T2, and T3; higher VCAM-1 at T1; higher TF at T2. No differences in concentrations were found in neonatal deaths. VPIs with hemodynamic or respiratory complications had higher Ang-2 at T3. Perinatal low Ang-1 and high Ang-2 associated with high VCAM-1 and TF in VPIs suggest a proinflammatory endothelial phenotype, resulting from the synergy of a pathological prenatal inheritance and a premature extrauterine transition.

## 1. Introduction

Premature birth is always the result of a pathological process, inflammatory or associated with dysfunctional placentation [1]. Hence, premature infants are at risk of multiple early complications, and neonatal outcomes are closely related to the gestational age (GA) at delivery [2].

The endothelium constitutes one of the most extensive internal surfaces of the body and can be considered conceptually a “systemically disseminated organ” [3]. It interacts with the soluble mediators of the inflammatory response and regulates vascular permeability, coagulation balance, and angiogenesis. Moreover, endothelium modulates the vascular tone and the transport of substances between the intra- and extravascular compartments [4].

The characterization of the endothelial profile at birth could provide valuable information about critical premature neonates and their antenatal history. Indeed, the maternal processes leading to prematurity are classified into endotypes involving different molecular pathways [5,6]. Therefore, different endotypes may determine specific endothelial phenotypes. Although neonatal and pregnancy diseases have shown in-vivo and ex-vivo alterations in numerous soluble endothelial markers, including adhesion molecules, tissue factor (TF), and endothelin 1 (ET-1), currently, little is known about the endothelial profile of preterm infants [7,8,9,10,11]. Understanding it in these patients is highly complex. The immature endothelium is subjected to continuous dynamic events during the fetal and postnatal period, influenced by gestational age, oxygen blood pressure, and placental vascular growth factors, according to the process named “endothelial developmental physiology” [12,13,14].

A family of endothelial growth factors, called angiopoietins, is gripping in this context. Initially discovered as regulators of angiogenesis, angiopoietins are competitors for the transmembrane tyrosine-protein kinase receptor Tie-2 (Tie2) expressed on the luminal endothelium [15]. Growing evidence has delineated the significant role of Angiopoietin-Tie2 related endothelial activation in acute lung injury (ALI), systemic inflammatory response syndrome (SIRS), and multiorgan dysfunction in adult patients and full-term newborns [16,17,18]. Furthermore, angiopoietin 1 (Ang-1) and angiopoietin 2 (Ang-2) play an antagonistic role in regulating the inflammatory response [3]. Indeed, plasma imbalance of Ang-2/Ang-1 ratio toward the proinflammatory Ang-2 has shown a predictivity for mortality in critically ill adult, and pediatric patients admitted to intensive care units [19].

Hence, to explore the role of these pathways in preterm birth, we designed a pilot trial on a cohort of very preterm infants (VPIs, <32 weeks of gestation). The study’s primary objective was to describe the profile of six soluble endothelial markers in the first 10 days of life and compare it with a control group of healthy full-term infants (FTIs). Secondary objectives were to evaluate the association between the profile of the endothelial markers and: (1) neonatal mortality; (2) early respiratory or hemodynamic complications. Our hypothesis was that VPIs had an unbalanced endothelial phenotype, whose trend is related to early outcomes.

## 2. Materials and Methods

This prospective observational pilot study was carried out from March to July 2021 at the III° level neonatal intensive care unit (NICU) of Fondazione IRCCS Ca’ Granda Ospedale Maggiore Policlinico, Milan, Italy. All procedures were carried out according to the principles of good clinical practice, the World Medical Association Helsinki Declaration, and the national legislative and administrative provisions in force. The study was approved by the local ethics committee (Milan Area 2, Italy) on 2 March 2021, with approval number 8997_2021. Furthermore, written informed consent to be part of the study and publication of the results was obtained from their parents.

### 2.1. Study Population

All consecutive VPIs were assessed at birth for eligibility. Exclusion criteria were: (1) prenatal hematological disorders, including intrauterine transfusions; (2) perinatal asphyxia, defined as an Apgar score at 10′ ≤ 5, or any pH ≤ 7.0 or base deficit ≥ 12 mmol/L in the first hour of life [20].

As control group, we evaluated all consecutive healthy FTIs (≥37 weeks of gestation) rooming-in hospitalized with indications to perform blood tests on the first day of life just for obstetric risk factors. FTIs with clinical signs suggestive of infection, respiratory distress, or hemodynamic instability at the sample test were excluded from the study.

In addition, as per department protocol, all VPIs received cardiac and cerebral ultrasound surveillance on days 1, 3, and 7. For both groups, we collected demographic data related to maternal medical history during pregnancy (multiple pregnancies, drugs, fetal growth, preeclampsia, chorioamnionitis), perinatal (gender, delivery mode, Apgar, cardiopulmonary resuscitation), and postnatal period (prematurity-related comorbidities, clinical course and outcome of blood tests in FTIs).

### 2.2. Soluble Endothelial Markers

The endothelial phenotype was characterized by measuring the serum concentration of six soluble molecules involved in vascular homeostasis: Ang-1, Ang-2, E-selectin, vascular adhesion molecule 1 (VCAM-1), TF, ET-1. Angiopoietins are also presented as the Ang-2/Ang-1 ratio.

The whole blood (0.5 mL) was collected in an ethylenediaminetetraacetic acid (EDTA)-containing test tube for each evaluation. The samples were centrifuged at 3000 rpm at 4 °C for 15 min. The supernatant was collected and stored at −20 °C. According to the manufacturer’s protocols, the analysis was performed with Luminex technology (R&D System, Inc., Minneapolis, MN, USA).

The sampling schedule was 1st (T1), 3rd (T2), and 7–10th (T3) day of life in the VPIs cohort and 1st day of life in the control group.

### 2.3. Prematurity-Related Outcomes

Neonatal mortality was defined as death within the 28th day of life. In addition, early respiratory and hemodynamic complications were evaluated as the occurrence in the first 72 h of any of the following: (1) Respiratory failure (need for mechanical ventilation); (2) Arterial hypotension (need for vasopressor drugs to maintain a mean arterial pressure ≥ 10th percentile for the patient’s GA) [21]; (3) ultrasound evaluation of hemodynamically significant patent ductus arteriosus (PDA) [22]; (4) ultrasound evaluation of severe (grades 3–4) intraventricular hemorrhage (IVH) [23]. Finally, a post hoc analysis of subgroups divided for maternal endotypes at T1 was performed to evaluate the prenatal contribution to the endothelial phenotype. The endotypes were classified as follows: inflammation (chorioamnionitis or placental abruption) vs. dysfunctional placentation (preeclampsia or fetal growth restriction) [6].

### 2.4. Statistical Analysis

According to the literature, we planned a sample size of 20 VPIs and 14 FTIs for this pilot study, allowing for the dropout of technical issues and adverse clinical outcomes [24]. Demographic characteristics of the two populations were presented as mean (SD), median [IQR], or n (%). Patients who have not completed the 3 scheduled samples are still included in the descriptive population assessments and comparing VPIs and FTIs. Mann-Whitney U test was used to compare FTIs and VPIs data, dead and survived infants’ values, and maternal endotype-related values. Friedman test with false discovery rate correction in post hoc tests was used to study significant variations between the 3 samples of each soluble marker; in particular, only VPIs with 3 available measurements were considered [25]. Linear regression models were used to study the association between GA or birth weight (independent variables) and soluble markers at T1 (dependent variables). Results are expressed as an estimated coefficient, 95% CI, and *p*-value. Statistical significance was defined for *p*-values < 0.05.

## 3. Results

### 3.1. Population

Baseline characteristics of the 20 VPIs and 14 FTIs enrolled are summarized in Table 1. In the VPIs group, 16 patients (80%) were found to be extremely preterm infants (GA < 28 weeks), and 15 (75%) were extremely low birth weight infants (ELBW, birth weight < 1000 g).

The neonatal mortality was 25%: 4 infants (20%) died in the first week of life, and 1 infant (5%) died on the 26th day of life. In this case, 14 infants (70%) reported at least 1 early respiratory or hemodynamic complication: 12 (60%) respiratory failure, 9 (45%) arterial hypotension, 10 (50%) hemodynamically significant PDA, 5 (25%) severe IVH. None of the infants presented an established diagnosis of early-onset sepsis. Only 6 (30%) VPIs did not show any early complications.

Here, 14 FTIs (100%) had normal physical examinations at birth. The obstetric indications for performing blood tests were: 9 cases (65%) of prolonged rupture of membranes (PROM, >18 h); 2 cases (14%) of maternal rectal-vaginal swabs positive for Streptococcus agalactiae with incomplete maternal prophylaxis [26,27]; 2 cases (14%) of brown amniotic fluid, 1 case (7%) of neglected pregnancy [28]. Blood test results showed negative c reactive protein (CRP) in 11 infants (79%). An increased CRP (>0.5 mg/dL) was observed in 2 PROM infants and the neglected pregnancy infant. The mean (SD) CRP in these 3 infants was 2.8 (1.6) mg/dL. None showed clinical signs of infection, respiratory distress, or hemodynamic instability throughout hospitalization; they did not perform intravenous antibiotic therapy and were not transferred to the NICU.

### 3.2. Endothelial Profile

The assessments of soluble endothelial markers were 67: 53 from the VPIs group and 14 from the control group. In this case 53 VPIs group observations are subdivided as follows: 20 (100%) at T1, 17 (85%) at T2, and 16 (80%) at T3. Defections are due to 4 deaths in the first week of life. ET-1 values were undetectable in all patients of both groups using the measurement technique. The complete results of the other soluble endothelial markers are summarized in Table 2.

#### 3.2.1. Relationship between VPIs and FTIs

**Angiopoietins**: VPIs compared with FTIs had:
Lower Ang-1 at T1 and T2 (*p* < 0.001);Higher Ang-2 at T1, T2, and T3 (*p* < 0.001);Higher Ang-2/Ang-1 ratio at T1, T2, and T3 (*p* < 0.001 at T1–T2; *p* = 0.017 at T3).**Adhesion molecules:** E-selectin was lower in VPIs at T1, T2, and T3 compared to the control group (*p* < 0.001 at T1–T2; *p* = 0.008 at T3). VCAM-1 showed higher values in VPIs at T1 than in FTIs (*p* < 0.001).**Tissue Factor:** TF was higher in the VPIs at T2 than in the control group (*p* < 0.001).

The relationship between soluble endothelial markers of VPIs and FTIs is described in Figure 1.

#### 3.2.2. Relationship between T1–T2–T3

Friedman test showed significant changes in the first 10 days of life in the soluble markers analyzed.

**Angiopoietins:** Ang-1 was higher at T3 compared to T1 [16.7 (14.9) vs. 5.4 (9.8) ng/mL, *p* = 0.013] and T2 [16.7 (14.9) vs. 4.1 (5.1) ng/mL, *p* = 0.005]; Ang-2 increased from T1 to T2 [27.3 (8.4) vs. 36.2 (9.5) ng/mL, *p* = 0.022], and it further increased comparing T1 and T3 [27.3 (8.4) vs. 34.8 (9.3) ng/mL, *p* = 0.013]; The Ang-2/Ang-1 ratio was lower at T3 compared to T1 [6.8 (8.9) vs. 11.1 (9.5) ng/mL, *p* = 0.034] and T2 [6.8 (8.9) vs. 15.4 (9.6) ng/mL, *p* < 0.001].**Adhesion molecules:** E-selectin was lower at T1 compared to T2 [39.1 (30.0) vs. 59.5 (32.5) ng/mL, *p* < 0.001] and T3 [39.1 (30.0) vs. 73.3 (35.1) ng/mL, *p* < 0.001]; VCAM-1 decreased from T1 to T2 [5244 (1226) vs. 3741 (1378) ng/mL, *p* = 0.005], subsequently remaining unchanged at T3.**Tissue Factor:** TF increased from T1 to T2 [0.098 (0.055) vs. 0.126 (0.050) ng/mL, *p* = 0.001], then it decreased from T2 to T3 [0.126 (0.050) vs. 0.093 (0.024) ng/mL, *p* = 0.010].

The relationship between soluble endothelial markers at T1-T2-T3 is described in Figure 2.

#### 3.2.3. Soluble Endothelial Markers and Baseline Parameters

Among the VPIs population, linear regression models showed negative associations between GA and Ang-2 and TF (Figure 3); the birth weight values resulted negative associated with TF (Coef. −0.0003, CI95% −0.0005; −0.0001, *p* = 0.006; Appendix A).

### 3.3. Endothelial Profile and Subgroups Analysis

#### 3.3.1. Neonatal Mortality

Mortality analysis was performed by comparing only the observations at T1 due to deaths before T2 collection of 3 out of 5 infants (60%). There were no significant differences in soluble makers between the 15 surviving and 5 dead VPIs (Table 3). The soluble markers showed the following mean values in dead compared with survivors VPIs: Ang-1 48% lower, Ang-2 24% lower, Ang-2/Ang-1 58% higher, E-selectin 23% higher, VCAM-1 16% higher, TF 139% higher. The dead patients showed significantly lower gestational age [25.5 (1.6) vs. 27.3 (1.3) weeks, *p* = 0.021] and weight [652 (89) vs. 945 (198) g, *p* = 0.005].

#### 3.3.2. Hemodynamic and Respiratory Complications

Testing for differences in VPIs with or without early complications, Ang-2 at T3 was significantly higher in the 14 infants with early complications [39 (9.2) vs. 27.7 (3.4) ng/mL, *p* = 0.005; Table 4]. The soluble markers showed the following mean values in complicated compared with uncomplicated VPIs at T1: Ang-1 65% higher, Ang-2 37% higher, Ang-2/Ang-1 ratio 100% higher, E-selectin 55% higher, VCAM-1 2% lower, TF 94% higher. Patients with complications showed significantly lower gestational age [26.3 (1.3) vs. 28 (1.3) weeks, *p* = 0.012] and birth weight [795 (189) vs. 1050 (182) g, *p* = 0.012].

#### 3.3.3. Maternal Endotypes

No significant differences were found in concentrations of soluble markers at T1 between 16 VPIs (80%) born from the inflammatory endotype vs. 4 VPIs (20%) born from the dysfunctional placentation endotype (Table 5). However, the soluble markers showed the following mean values in VPIs with inflammatory endotype compared with VPIs with dysfunctional placentation endotype: Ang-1 75% lower, Ang-2 19% higher, Ang-2/Ang-1 ratio 75% higher, E-selectin 89% higher, VCAM-1 15% higher, TF 11% higher.

## 4. Discussion

To our knowledge, this pilot study first explores in-vivo a perinatal endothelial profile of VPIs compared to a control group of FTIs. Previous studies have evaluated soluble angiogenic factors and markers of vascular contractility in preterm infants with bronchopulmonary dysplasia and hypoxic encephalopathy, respectively, but none of these assessed angiopoietins balance [11,29,30].

We observed significant differences in angiopoietins concentrations between VPIs and healthy FTIs, configuring a higher Ang-2/Ang-1 ratio from birth to the 10th day of life in the VPIs group. Ang-1 promotes endothelial quiescence by enhancing cell survival and downregulating gene expression of proinflammatory and pro-coagulating pathways [31]. Conversely, rapid endothelial exocytosis of Ang-2 during hypoxic or inflammatory endothelial injury increases the surface expression of cell adhesion molecules. This destabilizes intercellular junctions with a consequent increase in microvascular permeability, which is a key mechanism in the pathophysiology of organ damage [32]. According to these biological processes, the perinatal increase in Ang-2, associated with a reduction in the antagonistic effect of Ang-1, displays inflammatory endothelial activation in premature infants.

Angiopoietins also play a fundamental role in regulating fetal angiogenesis, and our finding of a negative association between GA and Ang-2 suggests a higher physiological expression in preterms. However, knockout mouse models for Ang-1 or Ang-2 demonstrated a fine prenatal regulation in the concentrations of Tie-2 receptor competitors [33,34]. Furthermore, the fetal absence of Ang-1, similarly to the overexpression of Ang-2, inevitably leads to abortion, demonstrating that Ang-1 plays the leading role in a correct and orderly angiogenic process while Ang-2 acts as its counter-regulatory [15]. Therefore, ontogenesis alone should not justify the imbalance of VPIs towards Ang-2. Considering the Ang-2 involvement in adult conditions common to preterms, such as SIRS and ALI, we could assume that the VPIs also manifest an inflamed endothelium [17,35].

Moreover, the inflammatory hypothesis is supported by the concurrent higher levels of VCAM-1 and TF in VPIs. Ex-vivo studies have shown that human umbilical vein endothelial cells (HUVECs) of ELBW infants can generate adequate up-regulation of adhesion molecules, including VCAM-1, in response to proinflammatory cytokines [36]. In-vivo studies confirmed this finding by increasing VCAM-1 on cord blood samples of preterms born from maternal chorioamnionitis [37]. Furthermore, the interleukin-1β stimulation of HUVECs increases the subendothelial exposition of TF through the induction of vascular permeability, which Ang-2 also stimulates [38]. In this regard, the association between GA and TF could be secondary to increased permeability induced by Ang-2 in the most immature patients.

Conversely, E-selectin values showed an opposite trend compared to VCAM-1 and TF. At birth, it was lower in VPIs but tended to progressively increase at T2 and T3 without reaching FTIs’ levels. This does not necessarily contradict the inflammatory hypothesis but encourages the theory of an immature expression of this protein in preterm infants [37]. Indeed, the capability of up-regulating E-selectin at very low GA is debated, and some experimental data exclude its expression before the 32nd week of GA [39]. Nevertheless, these results demonstrate that soluble E-selectin is already present at extremely low GA with a time-dependent up-regulation trend.

Two moments can induce an inflammatory endothelial phenotype in VPIs: the fetal life, depending on the maternal endotype, and the postnatal life, depending on the hemodynamic and respiratory adaptation. Understanding and quantifying the contribution of the two moments are challenging, but the exposure kinetics of the molecules involved can provide some inferences.

Ang-1 and VCAM-1 modulation depend on nuclear transcription factors and take up to 3 h to manifest [40,41,42]. This premise suggests that T1 sampling, performed shortly after birth, was only partially affected by postnatal factors and that intrauterine inheritance can play a key role in this finding. However, the origin of this inheritance is still to be explored. The relationship between inflammatory vs. dysfunctional placentation endotypes at T1 did not give detailed answers. Although there are no significant differences, the dysfunctional placentation endotype showed a consistently less proinflammatory phenotype for all parameters analyzed. Furthermore, as an additional prenatal factor, the possible effect of steroid prophylaxis (80% of our population) on the endothelial phenotype should be evaluated for future studies.

Considering the postnatal contribution to the endothelial phenotype, a strength of the study was the prospective enrollment of critically ill patients, 75% of whom were ELBW infants, with high complication rates. The Ang-2 increase at T2, with no decrease at T3, is significant in this setting. Furthermore, Ang-2 is stored in preformed stocks in Weibel-Palade bodies, allowing a concentration peak as early as 30 min after the inflammatory stimulus, associated with a marked reduction of its blood concentrations as early as 60 min [43]. Therefore, a postnatal stimulus crosses the first days of life of VPIs, maintaining a prevalent Ang-2 phenotype. This suggests that premature birth is a stressful event for the endothelium or probably that the endothelium reflects hemodynamic and respiratory stress. The late Ang-2 prevalence in VPIs suffering from early complications, associated with a proinflammatory trend already shown at T1 by this group, supported this relationship.

Conversely, the increase of Ang-1, associated with VCAM-1 and TF reduction, suggests an anti-inflammatory counter-regulation in the late transition phase. On the 10th day of life, hemodynamic and respiratory adaptation is a subacute process for infants who are not affected by severe complications, and this could correlate with a re-balancing of the Ang-2/Ang-1 ratio. Accordingly, a protective role of Ang-1 has been proposed, which would be consistent with the trend of Ang-1 shown by deceased newborns, even if not statistically significant. [17].

The findings of this study introduce new evidence on the perinatal endothelium of preterm infants, but some limitations require consideration. This is an exploratory pilot study, and the small sample size, especially of subgroups, only provides a trend that is limited in showing a predictive value of angiopoietins on mortality and early acute complications. In addition, by ethical practice, the control group is limited to a single observation performed on the first day of life and not to serial samplings as for the group of VPIs that would better show the postnatal trend in healthy term infants. Finally, the analytical method showed insufficient sensitivity to detect soluble ET-1 in both the VPIs and the control groups. The evaluation of vascular contractility-related markers remains an area yet to be explored in prematurity, as the few data available on the nitric oxide (NO), endothelial nitric oxide synthase (eNOS), and ET-1 levels have shown controversial results [10,44,45,46,47]. However, clinical evidence of impaired modulation in preterm infants’ systemic and pulmonary vascular contractility suggests further efforts.

In conclusion, exploring the endothelial profile in a cohort of premature infants revealed a perinatal proinflammatory attitude due to a reduced Ang-1 and an overexpressed Ang-2. The exact mechanisms responsible for this phenotype and a possible prognostic value of angiopoietins on mortality and early complications await further studies.

## Figures and Tables

**Figure 1 biomedicines-10-01185-f001:**
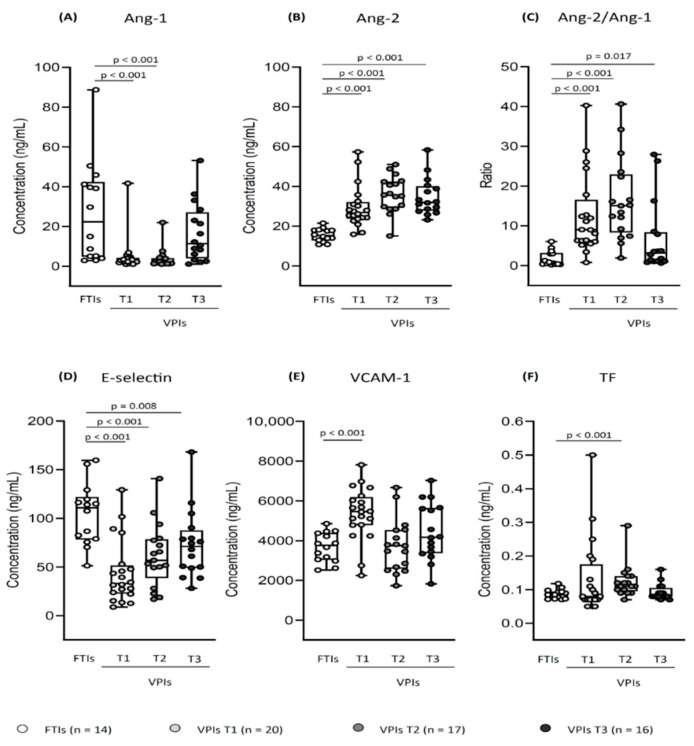
Soluble endothelial markers profile: relationship between very preterm infants (VPIs) at T1, T2, T3, and full-term infants (FTIs). Each dot represents a neonate, and each box whisker plot shows the median, interquartile range, and range; concentrations are expressed in ng/mL, Ang-2/Ang-1 is expressed as a ratio. (**A**) Ang-1 was lower in VPIs at T1 and T2 compared to FTIs (*p* < 0.001); (**B**) Ang-2 was higher in VPIs at T1, T2, and T3 compared to FTIs (*p* < 0.001); (**C**) Ang-2/Ang-1 ratio was higher in VPIs at T1, T2, and T3 compared to FTIs (*p* < 0.001 at T1–T2, *p* = 0.017 at T3); (**D**) E-Selectin was lower in VPIs at T1, T2, and T3 (*p* < 0.001 at T1–T2, *p =* 0.008 at T3); (**E**) VCAM-1 was higher in VPIs at T1 compared to FTIs (*p* < 0.001); (**F**) TF was higher in VPIs at T2 compared to FTIs (*p* < 0.001); Mann-Whitney U test. T1/T2/T3, 1st/3rd/7–10th day of life sample; Ang-1 and -2, angiopoietin-1 and -2; VCAM-1, vascular adhesion molecule-1; TF, tissue factor.

**Figure 2 biomedicines-10-01185-f002:**
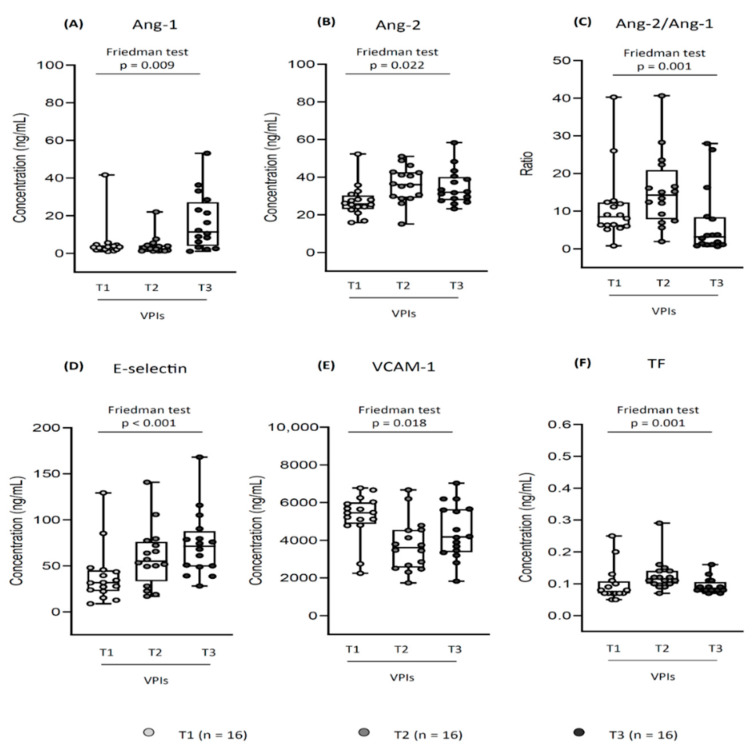
Soluble endothelial markers profile in very preterm infants (VPIs): relationship between T1, T2, and T3. Each dot represents a neonate, each box whisker plot shows the median, interquartile range and range; only infants with 3 available measurements were considered (n = 16); concentrations are expressed in ng/mL, Ang-2/Ang-1 is expressed as a ratio. (**A**) Ang-1 increased from T1 to T3 [5.4 (9.8) vs. 16.7 (14.9) ng/mL, *p* = 0.013], and from T2 to T3 [4.1 (5.1) vs. 16.7 (14.9), *p* = 0.005]; (**B**) Ang-2 increased from T1 to T2 [27.3 (8.4) vs. 36.2 (9.5), *p* = 0.022], and from T1 to T3 [27.3 (8.4) vs. 34.8 (9.3), *p* = 0.013]; (**C)** Ang-2/Ang-1 decreased from T1 to T3 [11.1 (9.5) vs. 6.8 (8.9) ng/mL, *p =* 0.034], and from T2 to T3 [15.4 (9.6) vs. 6.8 (8.9), *p* < 0.001]; (**D**) E-selectin increased from T1 to T2 [39.1 (30.0) vs. 59.5 (32.5), *p* < 0.001], and from T1 to T3 [39.1 (30.0) vs. 73.3 (35.1), *p* < 0.001]; (**E**) VCAM-1 decreased from T1 to T2 [5244 (1226) vs. 3741 (1378), *p* = 0.005]; (**F**) TF increased from T1 to T2 [0.098 (0.055) vs. 0.126 (0.050), *p* = 0.001], then decreased from T2 to T3 [0.126 (0.050) vs. 0.093 (0.024), *p* = 0.010]; the other relationships were not significant; Friedman test. T1/T2/T3, 1st/3rd/7–10th day of life sample; Ang-1 and -2, angiopoietin-1 and -2; VCAM-1, vascular adhesion molecule-1; TF, tissue factor.

**Figure 3 biomedicines-10-01185-f003:**
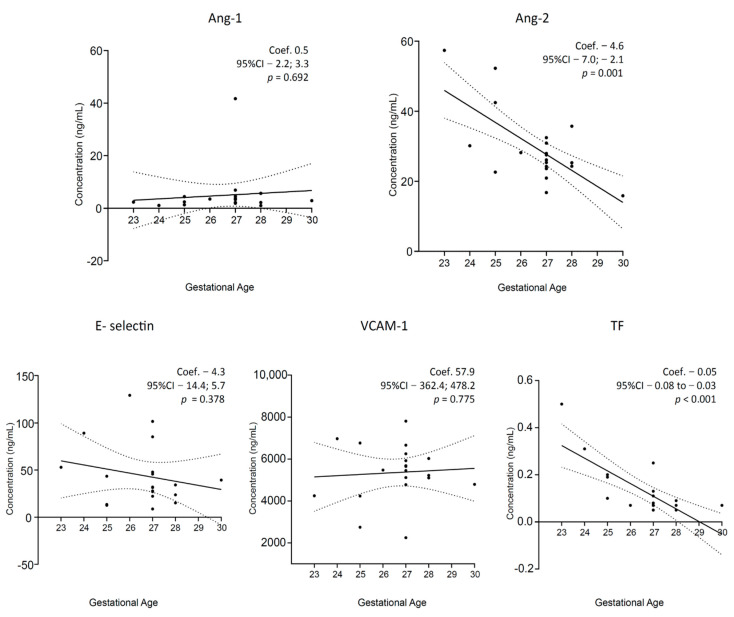
Soluble endothelial markers and gestational age in very preterm infants. Scatterplots of soluble markers at T1 are represented. A linear regression model was applied between gestational age and soluble markers. Concentrations are expressed as ng/mL. Coef, linear regression coefficient; p, *p*-values; CI, confident interval; T1, 1st day of life sample; Ang-1 and -2, angiopoietin-1 and -2; VCAM-1, vascular adhesion molecule-1; TF, tissue factor.

**Table 1 biomedicines-10-01185-t001:** Baseline characteristics of the study population.

**VPIs**	**Total (n = 20)**
Gestational age (weeks), mean (SD)	26.9 (1.5)
Birth weight (grams), mean (SD)	872 (218)
Male, n (%)	11 (55)
Multiple pregnancy, n (%)	8 (40)
Monochorionic, n (%)	2 (10)
Cesarean section, n (%)	18 (90)
Chorioamnionitis, n (%)	12 (60)
Placental abuption, n (%)	4 (20)
Pre-eclampsia/IUGR, n (%)	4 (20)
Antenatal steroids, n (%)	16 (80)
Apgar score 1’, median [IQR]	5 [4–6]
Apgar score 5’, median [IQR]	8 [7–8]
Cordonal ph (venous), mean (SD)	7.33 (0.09)
**FTIs**	**Total (n = 14)**
Gestational age (weeks), mean (SD)	39.6 (1.2)
Birth weight (grams), mean (SD)	3323 (257)
Male, n (%)	6 (43)
Multiple pregnancy, n (%)	0 (0)
Cesarean section, n (%)	3 (21)
Apgar score 1’, median [IQR]	9 [9–9]
Apgar score 5’, median [IQR]	10 [10–10]

VPIs, very preterm infants; SD, standard deviation; n, number; IQR, interquartile range; IUGR, intrauterine growth restriction; FTIs, full-term infants.

**Table 2 biomedicines-10-01185-t002:** Soluble endothelial markers profile.

Variables ^1^	VPIs T1 (n = 20)	VPIs T2 (n = 17)	VPIs T3 (n = 16)	FTIs (n = 14)	*p*-Value ^2^		
					(T1-FTIs)	(T2-FTIs)	(T3-FTIs)
Ang-1 (ng/mL)	4.9 (8.8)	3.9 (5)	16.7 (14.9)	27.0 (25.4)	<0.001	<0.001	0.313
Ang-2 (ng/mL)	29.5 (10.6)	36.1 (9.2)	34.8 (9.3)	15.5 (3.3)	<0.001	<0.001	<0.001
Ang-2/Ang-1 (ratio)	12.6 (10)	16.5 (10.4)	6.8 (8.9)	1.8 (1.9)	<0.001	<0.001	0.017
E-selectin (ng/mL)	44.1 (32.7)	61.6 (32.5)	73.2 (35.1)	104.5 (31.8)	<0.001	<0.001	0.008
VCAM-1 (ng/mL)	5359 (1335)	3745 (1334)	4456 (1441)	3680 (743)	<0.001	0.830	0.131
TF (ng/mL)	0.131 (0.112)	0.123 (0.049)	0.093 (0.025)	0.087 (0.014)	0.993	<0.001	0.886

^1^ Variables are expressed as mean (SD); ^2^ Mann-Whitney U test. VPIs, very preterm infants; FTIs, full-term infants; T1/T2/T3, 1st/3rd/7–10th day of life sample; Ang-1 and -2, angiopoietin-1 and -2; VCAM-1, vascular adhesion molecule-1; TF, tissue factor.

**Table 3 biomedicines-10-01185-t003:** Soluble endothelial markers at T1 and neonatal mortality in very preterm infants.

Variables ^1^	Dead (n = 5)	Survived (n = 15)	*p*-Value ^2^
Ang-1 (ng/mL)	2.9 (2.3)	5.6 (10.1)	0.553
Ang-2 (ng/mL)	27.4 (8.7)	35.9 (14.0)	0.230
Ang-2/Ang-1 (ratio)	17.4 (10.0)	11.0 (9.9)	0.197
E-selectin (ng/mL)	41.1 (29.9)	53.2 (42.4)	0.395
VCAM-1 (ng/mL)	5987 (1646)	5149 (1207)	0.735
TF (ng/mL)	0.232 (0.178)	0.097 (0.056)	0.098

^1^ Variables are expressed as mean (SD); ^2^ Mann-Whitney U test. T1, 1st day of life sample; Ang-1 and -2, angiopoietin-1 and -2; VCAM-1, vascular adhesion molecule-1; TF, tissue factor.

**Table 4 biomedicines-10-01185-t004:** Soluble endothelial markers and early hemodynamic and respiratory complications in very preterm infants.

Variables ^1^	with Complications	without Complications	*p*-Value ^2^
	**T1 (n = 14 vs. 6)**		
Ang-1 (ng/mL)	5.6 (10.5)	3.4 (1.5)	0.659
Ang-2 (ng/mL)	32.1 (10.9)	23.5 (7.4)	0.076
Ang-2/Ang-1 (ratio)	14.8 (11.3)	7.4 (2.2)	0.179
E-selectin (ng/mL)	49.4 (37.5)	31.8 (11.7)	>0.999
VCAM-1 (ng/mL)	5319 (1580)	5451 (517)	0.659
TF (ng/mL)	0.153 (0.128)	0.079 (0.029)	0.207
	**T2 (n = 11 vs. 6)**		
Ang-1 (ng/mL)	4.6 (6.1)	2.6 (1.0)	0.961
Ang-2 (ng/mL)	38.0 (7.9)	32.6 (11.0)	0.462
Ang-2/Ang-1 (ratio)	18.3 (12.5)	13.2 (3.4)	0.591
E-selectin (ng/mL)	64.7 (38.9)	55.8 (17.1)	0.961
VCAM-1 (ng/mL)	3434 (1432)	4315 (996)	0.149
TF (ng/mL)	0.131 (0.060)	0.109 (0.018)	0.5249
	**T3 (n = 10 vs. 6)**		
Ang-1 (ng/mL)	17.4 (17.3)	15.5 (11.1)	0.875
Ang-2 (ng/mL)	39.0 (9.2)	27.7 (3.4)	0.005
Ang-2/Ang-1 (ratio)	8.9 (10.7)	3.2 (2.9)	0.713
E-selectin (ng/mL)	83.0 (40.9)	57.0 (13.2)	0.147
VCAM-1 (ng/mL)	4212 (1612)	4863 (110)	0.368
TF (ng/mL)	0.099 (0.030)	0.082 (0.003)	0.4923

^1^ Variables are expressed as mean (SD); ^2^ Mann-Whitney U test. T1/T2/T3, 1st/3rd/7–10th day of life sample; Ang-1 and -2, angiopoietin-1 and -2; VCAM-1, vascular adhesion molecule-1; TF, tissue factor.

**Table 5 biomedicines-10-01185-t005:** Soluble endothelial markers at T1 and maternal endotype in very preterm infants.

Variables ^1^	Inflammation (n = 4)	Dysfunctional Placentation (n = 16)	*p*-Value ^2^
Ang-1 (ng/mL)	3.1 (1.7)	12.2 (19.6)	0.873
Ang-2 (ng/mL)	30.5 (11.3)	25.6 (7.0)	0.807
Ang-2/Ang-1 (ratio)	13.8 (10.7)	7.9 (5.8)	0.484
E-selectin (ng/mL)	48.7 (34.7)	25.8 (12.7)	0.211
VCAM-1 (ng/mL)	5501 (1206)	4790 (1864)	0.494
TF (ng/mL)	0.133 (0.120)	0.120 (0.089)	0.915

^1^ Variables are expressed as mean (SD); ^2^ Mann-Whitney U test. T1, 1st day of life sample; Ang-1 and -2, angiopoietin-1 and -2; VCAM-1, vascular adhesion molecule-1; TF, tissue factor.

## Data Availability

The data presented in this study are available on request from the corresponding author.

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
