# Peer review of "Proinflammatory Endothelial Phenotype in Very Preterm Infants: A Pilot Study"

_biomedicines, 2022, doi:10.3390/biomedicines10051185_

Round 1

Reviewer 1 Report

This study examined the endothelial effect of prenatal inflammatory process in very preterm infants (VPIs). They measured the levels of several factors including Ang-1, Ang-2, E-selectin, VCAM-1, tissue factor (TF), and endothelin 1 (ET-1). Overall, the results are solid and provide valuable information to the community. The only concern I have is that the organization and writing of result part need to be improved.

Author Response

Dear Reviewer, thanks for your suggestion.

Following your advice:

1) We have reorganized the results section as follows:

- Line [178]: we reversed points 3.2.1 and 3.2.2, putting first the relationship between VPIs and FTIs, then the T1-T2-T3 relationship. Figure 1 has become number 2, and vice versa.

- Line [188]: we have moved the paragraph with linear regressions and the related figure 3 into a separate paragraph 3.2.3 to line [229].

- Line [248]: we have moved table 3 to the end of paragraph 3.3.1.

- Line [255]: we added "at T1", which was missing.

- Line [259]: now table 4 does not split over 2 pages and is therefore more clearly readable.

2) We performed a linguistic review, with minor changes highlighted in the text, respectively:

- Line [25] “its predictive value”

- Line [44] “The endothelium constitutes one of the most extensive internal surfaces of the body and can be considered conceptually a “systemically disseminated organ”

- Line [153] “VPIs did not show any early complications”

- Line [209] “relationship between very preterm infants…”

- Line [218] “relationship between T1, T2, and T3. Each dot represents a neonate…”

- Line [278] “none of these assessed angiopoietins balance”

- Line [285] “…cell adhesion molecules. This destabilizes intercellular junctions…”

- Line [304] “In-vivo studies confirmed this finding by increasing VCAM-1…”

- Line [328] “…this inheritance is still to be explored. The relationship between…”

- Line [356] “that is limited in showing a predictive value of angiopoietins on mortality”

Reviewer 2 Report

This study "Proinflammatory endothelial phenotype in very preterm infants: a pilot study" is very interesting and well written.

In this pilot study, all procedures were carried out according to the principles of good clinical practice, the World Medical Association Helsinki Declaration, and the national legislative and administrative provisions in force. The study was approved by the local ethics committee (Milan Area 2, Italy) with approval number 8997_2021.

The authors concluded that very preterm infants are exposed to proinflammatory endothelial phenotype due to a reduced Ang-1 and an overexpressed Ang-2. The exact mechanisms are responsible for this phenotype not clear enough.

The results are clearly shown in the Tables and Figures. The authors used adequate statistical tests.

My opinion is that this manuscript is suitable for publication.

Author Response

Dear Reviewer, thanks for your review and your positive opinion.

Reviewer 3 Report

  1. All in all, it is a good article. The statistical part of the article is confusing. The number of subjects in the article is suitable for nonparametric statistics. In which comparison did the authors use the independent t-test? VPI and FTI data should be compared in pairs with the Mann-Whitney U test. VPI group baseline and later values should be compared with the Wilcoxon test. Dead and live infant values should be compared with the Mann-Whitney U test. Here, the independent t-test and ANOVA test are not suitable for this study.
  2. Why was not NO studied besides the existing biomarkers in this study?

Author Response

Dear Reviewer, thanks for your suggestions and questions.

1) Following your advice, we have reorganized the statistical part of the article in order to make it clearer.

- First of all, in the submitted manuscript, we wrongly reported the use of independent t-test, but actually, we did not use it, and we modified that methods section accordingly: lines [131, 133].

- We performed Friedman test instead of repeated measures ANOVA when comparing soluble endothelial markers profiles in the three-time points T1, T2, and T3, with Benjamini et al. correction for post hoc analysis [Ref: Benjamini, Y., Krieger, A. M. & Yekutieli, D. Adaptive linear step-up procedures that control the false discovery rate. Biometrika 93, 491–507 (2006)].

We modified the methods section accounting for this: line [135].

- Subsequently, we modified the result section 3.2.2, line [182], figure 2 and figure 2 legend, line [218], accounting for the Friedman test results: the statistical significance did not change, but six p-values varied slightly: Ang-1 T2 vs. T3, line [193]; Ang-2 T1 vs. T2 and T2 vs. T3 lines [194-195]; VCAM-1 T1 vs. T2, line [200]; TF T1 vs. T2 and T2 vs. T3, line [202, 203]. In addition, two new relationships have become significant and have been added: Ang-1 T1 vs. T3, line [192]; Ang-2/Ang-1 ratio T1 vs. T3, line [195].

2) The idea of measuring nitric oxide is very interesting, and we evaluated it in the study design. Technically for this pilot study, the NO dosage was difficult as it required additional kits to those already used, with additional volumes of blood, and in these very small patients, it was a problem. We are currently evaluating the best technique for its dosage, with good sensitivity and little volume required, and certainly, the study of the NO pathway will be the subject of our future studies.

Round 2

Reviewer 3 Report

I have no further comments